# *VivesDebate*: A New Annotated Multilingual Corpus of Argumentation in a Debate Tournament

Ramon Ruiz-Dolz [1,*], Montserrat Nofre [2], Mariona Taulé [2], Stella Heras [1] and Ana García-Fornes [1]

1    Valencian Research Institute for Artificial Intelligence (VRAIN), Universitat Politècnica de València, 46022 València, Spain; stehebar@upv.es (S.H.); agarcia@dsic.upv.es (A.G.-F.)
2    Centre de Llenguatge i Computació (CLiC-UB), Universitat de Barcelona, 08007 Barcelona, Spain; montsenofre@ub.edu (M.N.); mtaule@ub.edu (M.T.)
*    Correspondence: raruidol@dsic.upv.es

**Abstract:** The application of the latest Natural Language Processing breakthroughs in computational argumentation has shown promising results, which have raised the interest in this area of research. However, the available corpora with argumentative annotations are often limited to a very specific purpose or are not of adequate size to take advantage of state-of-the-art deep learning techniques (e.g., deep neural networks). In this paper, we present *VivesDebate*, a large, richly annotated and versatile professional debate corpus for computational argumentation research. The corpus has been created from 29 transcripts of a debate tournament in Catalan and has been machine-translated into Spanish and English. The annotation contains argumentative propositions, argumentative relations, debate interactions and professional evaluations of the arguments and argumentation. The presented corpus can be useful for research on a heterogeneous set of computational argumentation underlying tasks such as Argument Mining, Argument Analysis, Argument Evaluation or Argument Generation, among others. All this makes *VivesDebate* a valuable resource for computational argumentation research within the context of massive corpora aimed at Natural Language Processing tasks.

**Keywords:** argumentation; corpus; debate; Natural Language Processing; Argument Mining; Argument Analysis; Argument Evaluation; Argument Generation

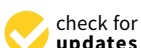



## 1. Introduction

Argumentation is the process by which humans reason to support an idea, an action or a decision. During this process, arguments are used by humans to shape their reasoning using natural language. In an attempt to understand how human reasoning really works, researchers have focused on analysing and modelling the use of arguments in argumentation. This problem has been usually approached by philosophers and linguists [1–3]. However, recent advances in Artificial Intelligence (AI) show promising results in Natural Language Processing (NLP) tasks that were previously unfeasible (e.g., machine translation, text summarisation, natural language generation), but now leave the door open to exploring more complex aspects of human language and reasoning. Computational argumentation is the area of AI that aims to model the complete human argumentative process [4,5], and encompasses different independent tasks that address each of the main aspects of the human argumentation process (Figure 1). First, Argument Mining [6,7] focuses on the automatic identification of arguments and their argumentative relations from a given natural language input. Second, argument representation [8–11] studies the best computational representations of argument structures and argumentative situations in different domains. Third, argument solving (or evaluation) [8,12–14] researches methods and algorithms to automatically determine the set of *acceptable* (i.e., winner) arguments from the complete set of computationally represented arguments. Finally, the Argument Generation [15–18] task is mainly focused on the automatic creation of new arguments from a context and a set of known information regarding some specific topic.

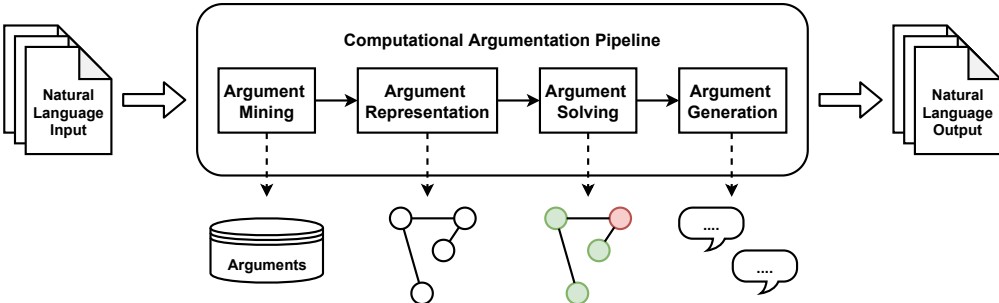

**Figure 1.** General pipeline for Computational Argumentation tasks.

Due to the huge heterogeneity of the tasks, each one of them requires different corpus structures and annotations to be approached from the computational viewpoint. Thus, depending on the data source, the annotations, and the task, a corpus might only be useful to approach a unique aspect of argumentation. For example, a simple corpus with small unrelated pieces of text annotated with *argument/non-argument* labels will only be useful for the Argument Mining task. In addition to this limitation, there is the great complexity underlying the human annotation of such corpora. This elevated complexity has a devastating impact on the versatility of the available corpora. The majority of the identified publicly available data for computational argumentation research is annotated either for a very specific task of the complete argumentation process, and does only consider its most superficial aspects (see Section 5). Furthermore, Deep Learning (DL) [19] has recently shown outstanding results in many different AI areas (e.g., NLP and Computer Vision among others). DL differs from the previous classic machine learning approaches in a major aspect—the data representations. While classic machine learning approaches usually required an important effort in feature engineering the input for each specific task, DL algorithms model the representation of each input automatically during the training process. However, despite presenting significantly superior results, DL approaches require a large amount of data to observe this improvement in the experimentation. Computational argumentation research has recently focused on the implementation of DL algorithms to approach each of its underlying tasks. In Argument Mining, the Transformer architecture [20] that presented outstanding results in the majority of NLP tasks has become the focus of attention. Recent research compares and proposes new Transformer-based neural architectures for both Argument Mining [21] and argument relation identification [22]. In argumentation solving, recent research proposes a deep graph neural network to automatically infer the *acceptable* arguments from an argumentation graph [14]. Finally, the latest Argument Generation research proposals explore the use of DL architectures to automatically generate natural language arguments [18,23] rather than using templates or retrieving arguments from a database. Thus, this trend of applying, adapting and proposing new state-of-the-art approaches to computational argumentation research makes the creation of new high-quality large corpora a priority. From the publicly available resources for computational argumentation research, it is possible to observe a strong trade-off between the size of the corpora and the *quality* of the annotations. We understand the term *quality* in this situation as the depth that annotations present from an argumentative viewpoint. The majority of the most extensive corpora available for computational argumentation research are usually focused on a very specific argumentative concept (e.g., segmentation, argument component identification, etc.), and only consider short pieces of argumentative text, which makes it possible to simplify (or even automate) the annotation process. However, these simplifications imply a significant loss of context and information from the annotated arguments.

The main objective of this article is to present *VivesDebate*, a new annotated argumentative multilingual corpus from debate tournaments. For that purpose, the contribution of this paper is threefold: (i) the creation of a new resource for computational argumentation research; (ii) the description of the annotation guidelines followed in the creation of the corpus; and (iii) the comparison and review of ten of the most relevant corpora for compu-

tational argumentation research. The *VivesDebate* corpus has been created based on three main aspects that are of paramount importance in recent developments in AI and computational argumentation: the size, the quality, and the versatility provided by the corpus in its different possible uses. The *VivesDebate* corpus has a total of *139,756* words from 29 annotated debates from the 2019 university debate tournament organised by the *"Xarxa Vives d'universitats"* (https://www.vives.org/programes/estudiants/lliga-debat-universitaria/, accessed on 2 August 2021). Each debate is annotated in its complete form, making it possible to keep the complete structure of the arguments raised in the course of the debate. Thus, the presented corpus is a relevant contribution to most of the main computational argumentation tasks such as Argument Mining, argument representation and analysis, argument solving, and Argument Generation and summarisation. Furthermore, the debates have been machine-translated from their original language (i.e., Catalan) to Spanish and English languages. The *VivesDebate* corpus is released under a Creative Commons Attribution-NonCommercial-ShareAlike 4.0 International license (CC BY-NC-SA 4.0) and can be freely downloaded from Zenodo (https://doi.org/10.5281/zenodo.5145656, accessed on 2 August 2021).

The rest of the paper is structured as follows. Section 2 defines professional debate tournaments and their structure, from which our corpus has been created. Section 3 thoroughly describes the methodology followed during the annotation process. Section 4 analyses the *VivesDebate* corpus and presents its most relevant features. Section 5 analyses and compares the most important existing corpora for computational argumentation research. Finally, Section 6 highlights the main conclusions and the future research that will take this corpus as its starting point.

## 2. Argumentation in Professional Debate Tournaments

Debate tournaments and competitions exist in different forms. Each type of debate has its own rules, structure, and aim. However, regardless of these differences, in every type of debate the winner is always the one presenting the best arguments and the most solid reasoning. Thus, given this condition, it is natural to think that one of the best sources to analyse the human argumentative discourse are debate competitions, mainly due to the higher quality of the reasoning presented by the participants. In this section, we thoroughly describe the standard academic debate tournament, which has served as the source of the corpus presented in this paper. This type of debate presents one of the most popular structures and rules used in university debate tournaments. First, a controversial topic is chosen, and the debating question is proposed in a way that two conflicting stances are created (in favour or against). Each debate is divided into three main phases: the introduction, the argumentation and the conclusion. Each team, consisting of three to five debaters (university students), is randomly assigned a stance for the tournament topic at the beginning of each debate. The team opening the debate is also drawn before its start. In the subsequent description of the flow of the debate, we will assume that the proposing (in favour) team begins, and the opposing (against) team follows up. Thus, the proposing team opens the debate with a 4 min introduction, where the main aspects that will be used to support their arguments are presented. Then, the opposing team is able to introduce their main ideas on the topic in another 4 min introduction. Once the introduction phase concludes, each team has two rounds of 6 min to argue their stances by presenting new arguments or supporting the previously introduced ones. Furthermore, in the argumentation phase, participants can also attack the arguments proposed by the other team. Finally, the debate is closed by each team's 4 min conclusion. The order in which each team concludes its argumentation is inverted with respect to the previous phases of the debate. Thus, in our instance of a debate where the proposing team's introduction was the first phase, the opposing team will be the first to present its main conclusions, and the debate will be closed with the conclusion of the proposing team. Figure 2 summarises the presented structure of academic debates from which the *VivesDebate* corpus has been created.

| Introduction (INTRO) 8 min | Proponent team: Presentation of its main lines of argument | 4 min |
|---|---|---|
| | Opponent team: Presentation of its main lines of argument | 4 min |
| Argumentation (ARG1) 12 min | Proponent team: Development of its main arguments | 6 min |
| | Opponent team: Development of its main arguments and rebuttal | 6 min |
| Argumentation (ARG2) 12 min | Proponent team: Reinforcement of its main arguments and rebuttal | 6 min |
| | Opponent team: Reinforcement of its main arguments and rebuttal | 6 min |
| Conclusion (CONC) 8 min | Opponent team: Conclusion of the debate | 4 min |
| | Proponent team: Conclusion of the debate | 4 min |

**Figure 2.** General structure for academic debate tournaments.

The outcome of each debate is decided by a Jury that evaluates six different aspects of the debate weighted by their relevance. First, the Jury assesses how solid each team's thesis is and how it has been defended during the debate (22.5%). Second, the Jury evaluates essential aspects of the argumentation such as the relation of the arguments to the topic, the strength and originality of the presented arguments, and the coherence of the discourse and its structure (22.5%). Third, the Jury assesses how well each team has reacted and adapted to the adversary's attacks and arguments (20%). Fourth, the Jury evaluates the security in discourse and the capacity of finding weak spots in the adversary's argumentation (15%). Fifth, aspects such as oral fluency, semantic and grammatical correctness, the richness of the vocabulary used, and non-verbal language are also assessed by the Jury (10%). Finally, the Jury considers positively the respectful attitude shown during the debate (10%).

A numerical score is assigned to each one of these aspects during the final deliberation, and a weighted sum of the six values indicates the score of each team. Furthermore, each team can be penalised if some specific conditions are met during the debate. Three different penalisation degrees are considered depending on their severity: warnings, minor faults, and serious faults. The warnings do not have a direct impact on the previously defined score, and may happen when the team members talk between themselves during the debate, and when the speakers do not comply with the assigned duration of each phase of the debate. A minor fault will reduce the score by 0.5, and happens with the accumulation of two warnings, with minor behavioural issues, and when a team uses fake news to support their arguments. Finally, a serious fault reduces the score by three points, and will only happen if a team commits serious disrespectful acts (e.g., insults, racism, misogyny, etc.), or violates the rules of the tournament. Each Jury is specifically constituted for each debate and is composed of at least three members that are assigned before starting the debate. Thus, the final score (*FS*, Equation (1)) consists of a normalisation of the score (*S*) minus the penalisation (*P*) assessed by each member of the Jury ($\forall j \in J$):

$$FS = \frac{\sum_j^J S_j - P_j}{|J|}. \tag{1}$$

## 3. Annotation Methodology

In this section, we describe the annotation tagset used, the criteria applied and the annotation process carried out, including the Inter-Annotator Agreement tests conducted for the annotation of the *VivesDebate* corpus. The annotation task consists of three main subtasks: first, the annotators review and correct the transcriptions automatically obtained by the MLLP transcription system (https://ttp.mllp.upv.es/, accessed on 2 August 2021) [24],

the IberSpeech-RTVE 2020 TV Speech-to-Text Challenge award winning transcription system developed by the Machine Learning and Language Processing (MLLP) research group of the VRAIN. Then, the Argumentative Discourse Units (ADUs) of each debate, which are the minimal units of analysis containing argumentative information, are identified and segmented. Finally, the different types of argumentative relationships between the previously identified ADUs are annotated. All these tasks are manually carried out by two different annotators and supervised by a third senior annotator. In the following, we describe in more detail each of these subtasks.

### 3.1. Revision and Correction of Automatic Transcriptions

The first task we performed was the revision of the automatic transcriptions of each debate. The duration of each debate was approximately 50 min. The annotators reviewed whether these automatic transcriptions corresponded to the audio recorded in the videos. Due to the time required to completely correct all the transcriptions, we decided to focus on the quality of the transcriptions corresponding to the ADUs in order to ensure the comprehensibility of the arguments presented during the debate and avoid misunderstandings, as a general criterion. The remaining text, which will not be part of the ADUs, was checked for inconsistencies or glaringly obvious errors. Regarding the specific criteria established, we agreed:

1.  To maintain the transcriptions of the different linguistic variants of the same language used in the original debates—Balearic, central and north-western Catalan and Valencian—as well as the use of words or expressions from other languages, but which are not normative, such as borrowings from Spanish and from English. It is worth noting that the eastern variants (Balearic and central Catalan) are the prevailing variants of the MLLP transcriber.
2.  To correct spelling errors, such as '*autonoma' instead of 'autònoma/autonomous' (missing accent), '*penalitzar vos' instead of 'penalitzar-vos/penalize you' (missing hyphen).
3.  To amend those words that were not correctly interpreted by the automatic transcriber, especially wrongly segmented words. For instance, '*debatre/to debate' instead of 'debatrà/he or she will debate', or '*desig de separa/the desire to (he/she) separates' instead of 'desig de ser pare/the desire to be a father'. In the first example, the automatic transcription does not correctly interpret the tense and person of the verb, and in the second example it wrongly interprets as a single word ('separa') two different words 'ser pare', probably due to the elision of 'r' when we pronounce 'ser' and due to the confusion that can be caused by the Catalan unstressed vowels 'a' and 'e', which in some linguistic variants are pronounced the same way. Most of these errors are related to homophonous words or segments, which the automatic transcriber cannot distinguish correctly, and are also probably due to the different linguistic variants of the same language used in the debates (Balearic, central and north-western Catalan and Valencian).
4.  To follow the criteria established by the linguistic portal of the Catalan Audiovisual Media Corporation (http://esadir.cat/, accessed on 2 August 2021) for spelling the names of persons, places, demonyms, and so forth.
5.  To capture and write down the main ideas in those cases in which the quality of the audio does not allow us to understand part of the message conveyed.
6.  Noisy sounds and hesitations (e.g., 'mmm', 'eeeeh'), self-corrections (e.g., 'mètode arlt alternatiu/arlt alternative method') and repetitions ('el que fa el que fa és ajudar/what it does it does is to help)' are not included because they do not provide relevant information for computational tasks focused on argumentation.

The data that are automatically transcribed and manually reviewed appear in plain text format without punctuation marks and without capital letters. The revision of the automatic transcriptions took us an average of two hours for each debate, even though we performed a superficial revision of those fragments of text in which ADUs were not present. Therefore, this type of revision is undoubtedly a time-consuming task.

### 3.2. Segmentation of Debates in Argumentative Discourse Units (ADUs)

The next task consisted of segmenting the text into ADUs (**ADU** tag) and annotating it, identifying: (a) the participant who uttered the ADU and the part of the debate in which it was used (i.e., in the introduction, argumentation or conclusion) by means of the **PHASE** tag; (b) their stance towards the topic of the debate (against or in favour) annotated with the **STANCE** tag; and the number of the argument presented during the argumentation part tagged as **ARGUMENT_NUMBER** (the identified arguments are labelled with a numerical value from 1 to the maximum number of arguments found in the debate) (see Section 2 and Figure 2 for more information). The preparatory part of the debate, in which the topic and the stance each team had to adopt were decided, and the conclusions were not segmented.

An Argumentative Discourse Unit (ADU) is defined as the minimal unit of analysis containing argumentative information [25]. Therefore, an argument can consist of one or more ADUs, each contributing a different (or complementary) argumentative function (e.g., premises, pieces of evidence, claims).

Next, we describe the criteria followed for the segmentation of ADUs and the tags assigned to each ADU for identifying them. This task was also manually performed by the same two annotators and was reviewed by the senior annotator.

Segmentation Criteria

Two general criteria were applied to the segmentation of ADUs: the first is that ADUs are created following the chronological order in which they appear in the discourse. Each ADU was assigned a unique **ID** tag for identifying them and showing their position in the chronological sequence. The second criterion is related to the quality of ADUs, which means that their content has to be clear, comprehensible and coherent. Therefore, this involves a further revision and, if necessary, a correction of the text. Each ADU corresponds to a transcribed text segment considered as a unit of argumentation and was included in the **ADU** tag. The ADUs are generally equivalent to a sentence or a dependent clause (for instance a subordinated or coordinated clause). It is worth noting that we also found ADUs which contained a subsegment that was, in turn, another ADU (for instance, relative clauses, as we will see below). In this case, the second ADU was also assigned its own ID and ADU tags.

The specific criteria followed for the segmentation of ADUs were the following:

- Punctuation marks must not be added to the content of the ADUs. The annotators could view the debate recording to solve ambiguous interpretations and avoid a misinterpretation of the message.
- Anaphoric references are left as they are, there is no need to reconstruct them, that is, the antecedent of the anaphora is not retrieved.

    (1a) [és una forma d'explotació de la **dona**]$_{ADU1}$
    [és una forma de cosificar-**la**]$_{ADU2}$
    [i és una forma que fa que vulneri la **seva** dignitat]$_{ADU3}$

    (1b) [it is a way of exploiting **women**]$_{ADU1}$
    [it is a way of objectifying **them**]$_{ADU2}$
    [and it is a way of violating **their** dignity]$_{ADU3}$

    In example (1), the text contains three different ADUs and two anaphoric elements appear in the second and third ADUs, '-la/her' in ADU2 and 'seva/her' in ADU3, which refer to the same entity ('dona/woman'), but we did not retrieve their antecedent in the corresponding ADUs.

- Discourse markers (2) must be removed from the content of the ADUs, but the discourse connectors (3) will be kept. Discourse connectors are relevant because they introduce propositions indicating cause, consequence, conditional relations, purpose, contrast, opposition, objection, and so forth, whereas discourse markers are used to introduce a topic to order, to emphasize, to exemplify, to conclude,

and so forth. We followed the distinction between discourse connectors and markers established in the list provided by the Language and Services and Resources at UPC (https://www.upc.edu/slt/ca/recursos-redaccio/criteris-linguistics/frases-lexic-paragraf/marcadors-i-connectors, accessed on 2 August 2021).

(2) Examples of discourse markers: 'Respecte de/ regarding'; 'en primer lloc/first or firstly'; 'per exemple/for instance'; 'en d'altres paraules/in a nutshell'; 'per concloure/in conclusion'.

(3) Examples of discourse connectors?: 'per culpa de/due to'; 'a causa de/because of'; 'ja que/since'; 'en conseqüència/consequently'; 'per tant/therefore'; 'si/if'; 'per tal de/in order to'; 'tanmateix/however'; 'encara que/although'; 'a continuació/then'.

- Regarding coordination and juxtaposition, we segmented coordinated sentences differently from coordinated phrases and words: (a) In coordinated sentences, each sentence was analysed as an independent ADU and the coordinating conjunction (e.g., copulative, disjunctive, adversative, distributive) was included at the beginning of the second sentence (4). The type of conjunction can be used to assign the argumentative relation in the following task; (b) In coordinated phrases and words, each of the joined elements are included in the same single ADU (5).

(4a) [l'adopció s'està quedant obsoleta]$_{ADU1}$
　　[**i** per això hem de legislar]$_{ADU2}$
(4b) [adoption is becoming obsolete]$_{ADU1}$
　　[**and** that's why we have to legislate]$_{ADU2}$
(5a) [**justícia i gratuïtat** per evitar la desigualtat social]$_{ADU1}$
(5b) [**justice and gratuity** to avoid social inequality]$_{ADU1}$

- Regarding subordinated sentences, the subordinated (or dependent) clause is analysed as an ADU that is independent from the main (or independent) clause, and includes the subordinating conjunction (6). The type of subordinating conjunction (e.g., causal, conditional, temporal, etc.) can be used to assign the argumentative relation.

(6a) [**si** s'acaba el xou]$_{ADU1}$ [s'acaba la publicitat]$_{ADU2}$
(6b) [**if** the show is over]$_{ADU1}$ [advertising is over]$_{ADU2}$

In example 6, two different ADUs are created, in which the second clause (ADU2) will then be annotated as an inference argumentative relation from the first clause (ADU1). In this way, if later, there a proposition appears that is only related to one of the two previous clauses, this proposition can be related to the corresponding ADU.

- Regarding relative clauses, the relative clause is included in the same ADU as the main clause, because these clauses function syntactically as adjectives. However, they can be treated as a subsegment of the ADU if the relative clause acts as an argument (7).

(7a) [suposa un desig de les persones que fa perpetuar un rol històric de la dona]$_{ADU1}$
　　[que fa perpetuar un rol històric de la dona]$_{ADU2}$
　　[afirma i legitima que les dones han de patir]$_{ADU3}$
(7b) [this presupposes a desire by people to perpetuate a historical role for a woman]$_{ADU1}$
　　[to perpetuate a historical role for a woman]$_{ADU2}$
　　[this asserts and legitimatizes the idea that women must suffer]$_{ADU3}$

In (7), the relative clause is segmented as an independent ADU2, because it is the argument to which ADU3 refers.

- In reported speech or epistemic expressions, we distinguish whether the epistemic expression is generated by one of the participants in the debate (8) or is generated by another (usually well-known or renowned) person (9). In the former, the subordinate clause is only analysed as an ADU while, in the latter, the whole sentence is included in the same ADU.

(8a) Jo pense que es deurien prohibir les festes amb bous ja que impliquen maltractament animal

[es deurien prohibir les festes amb bous]$_{ADU1}$
[ja que impliquen maltractament animal]$_{ADU2}$
(8b)  I think bullfights should be banned as they involve animal abuse
[bullfights should be banned as they involve animal abuse]$_{ADU1}$
[as they involve animal abuse]$_{ADU2}$
(9a)  [Descartes pensa que cos i ànima són dues entitats totalment separades]$_{ADU1}$
(9b)  [Descartes thinks that body and soul are two totally separate entities]$_{ADU1}$

In example (8), the ADU already indicates which specific participant uttered this argument in the STANCE and ARGUMENT_NUMBER tags associated, as we describe in more detail below (Section 4.2). Therefore, including this information would be redundant. However, in example (9), it would not be redundant and could be used in further proposals to identify, for instance, arguments from popular, well-known or expert opinion and arguments from witness testimony.

- With regard to interruptions within the argumentative speech produced by the same participant, the inserted text will be deleted (10), whereas if the interruption is made by a participant of the opposing group, it will be added to another ADU (11).

(10a)  el que estan fent vostès, i aquest és l'últim punt, és culpabilitzar a la víctima
[el que estan fent vostès és culpabilitzar a la víctima]$_{ADU1}$
(10b)  what you are doing, and this is my last point, is to blame the victim
[what you are doing is to blame the victim]$_{ADU1}$
(11a)  centenars de dones han firmat un manifest per tal de garantir d'adherir-se a la seua voluntat de ser solidàries, *que passa si els pares d'intenció rebutgen el nen i on quedaria la protecció del menor en el seu model*, completament garantida per l'estat
[centenars de dones han firmat un manifest per tal de garantir d'adherir-se a la seua voluntat de ser solidàries completament garantida per l'estat]$_{ADU1}$
[que passa si els pares d'intenció rebutgen el nen i on quedaría la protecció del menor en el seu model]$_{ADU2}$
(11b)  hundreds of women have signed a manifesto to ensure that they adhere to their willingness to be in solidarity, *what happens if the intended parents reject the child and where would the protection of the child be in your model*, fully guaranteed by the state
[hundreds of women have signed a manifesto to ensure that they adhere to the to their willingness to be in solidarity fully guaranteed by the state]$_{ADU1}$
[what happens if the intended parents reject the child and where would the protection of the child be in your model]$_{ADU2}$

In (11), the initial fragment of text is segmented into two different ADUs. The interruption (in italics) is segmented separately and tagged as ADU2. If an argumentative relation is observed in an interruption made by the same participant, this part of the text will be analysed as a new ADU and, therefore, will not be removed since it establishes the relationship between arguments.

- Interrogative sentences are analysed as ADUs because they can be used to support an argument (12), except when they are generic questions (13).

(12a)  [Això fa que no necessàriament ho valori econòmicament?]$_{ADU1}$ [No]$_{ADU2}$
(12b)  [Does that necessarily mean that I do not value it economically?]$_{ADU1}$ [No]$_{ADU2}$
(13a)  Què en penses d'això?
(13b)  What do you think about that?

It should be noted that tag questions are not annotated as ADUs. In (14) 'oi?/right?' is not tagged as an ADU.

(14a)  [Això fa que no necessàriament ho valori econòmicament]$_{ADU1}$ oi?
(14a)  [Does that necessarily mean that I do not value it economically]$_{ADU1}$ right?

- In the case of emphatic expressions (15), only the main segment is included in the ADU.

(15a)  sí que [hi ha la possibilitat]$_{ADU1}$
(15b)  yes [there is the possibility]$_{ADU1}$

- Examples and metaphoric expressions are annotated as a single ADU, because the relationship with another ADU is usually established with the whole example or the whole metaphor (16). In cases in which the relationship with another ADU only occurs with a part of the metaphorical expression or example, a subsegment can be created with its corresponding identity ADU.

  (16a) [aquest mateix any una dona es va haver de suïcidar just abans del seu desnonament o per exemple una mare va saltar per un pont amb el seu fill perquè no podia fer-se càrrec d'un crèdit bancari]$_{ADU1}$
  [si la gent és capaç de suïcidar-se per l'opressió dels diners com no es vendran a la gestació subrogada]$_{ADU2}$

  (16a) [this same year a woman had to commit suicide just before her eviction or for example a mother jumped off a bridge with her child because she could not pay back a bank loan]$_{ADU1}$
  [if people are able to commit suicide because of the oppression of money why shouldn't they sell themselves in surrogacy]$_{ADU2}$

  The ADU2, in (16), which includes an example, is related to the previous ADU1.

- Expressions including desideratum verbs (17) are not considered ADUs.

  (17a) A mi m'agradaria anar a l'ONU i explicar els mateixos arguments per a que aquesta prohibició no sigui només a Espanya

  (17b) I would like to go to the UN and present the same arguments so that this prohibition is not only in Spain.

*3.3. Annotation of Argumentative Relationships between ADUs*

Once the ADUs are identified and segmented, the aim of the following task is to establish the argumentative relationships between ADUs and to annotate the type of relation held. We use the **RELATED_ID** (**REL_ID**) and **RELATION_TYPE** (**REL_TYPE**) tags for indicating these argumentative relationships. The REL_ID tag is used to indicate that an ADU2 holds an argumentative relationship with a previous ADU1 (18). The ID identifies the corresponding ADUs. It is worth noting that the relationships between ADUs almost always point to previous ADUs, following the logic of discourse, and that not all the ADUs have argumentative relations with other ADUs. There are cases in which several ADUs maintain a relationship with a single previous ADU, and all of them are indicated. An ADU may be related to more than one previous ADU, but the type of relationship with each of them is different. In these cases, we annotated the REL_ID and REL_TYPE for each different type of relationship generated from the same ADU. The annotation of the argumentative relations mainly occurs in the argumentation phase of the debate, but may also appear in the introduction phase.

Next, we describe the three types of argumentative relationships—inference, conflict and reformulation—which represent different semantic relations between two propositions. These relationships are annotated with the REL_TYPE tag and the corresponding values are **RA** for inference, **CA** for conflict and **MA** for reformulation. The notation used for the argumentative relations has been adopted from the Inference Anchoring Theory (IAT) [26] paradigm in order to provide a coherent labelling with previous corpora.

- Inference (RA) indicates that the meaning of an ADU can be inferred, entailed or deduced from a previous ADU (18). As already indicated, the direction of the inference almost always goes from one ADU to a previous ADU, but we have also found cases in which the direction is the opposite, that is, the inference goes from a previous ADU to a following one, although there are fewer cases (19). Therefore, inference is a meaning relation in which the direction of the relationship between ADUs is relevant, and this direction is represented by the REL_ID tag.

  (18a) [la gestació subrogada és una pràctica patriarcal]$_{ADU1}$
  [ja que el major beneficiari d'aquesta pràctica és l'home]$_{ADU2}$ **REL_ID=1 REL_TYPE=RA**

(18b) [surrogacy is a patriarchal practice]$_{ADU1}$
   [since the main beneficiary of this practice is the man]$_{ADU2}$ **REL_ID=1 REL_TYPE=RA**
(19a) [no tot progrés científic implica un progrés social]$_{ADU1}$ **REL_ID=2;3 REL_TYPE=RA**
   [l'energia nuclear és la mare de la bomba atòmica]$_{ADU2}$
   [Els pesticides que multiplicaven les collites han estat prohibits per convertir el
   aliments en insalubres]$_{ADU3}$
(19b) [not all scientific progress implies social progress]$_{ADU1}$ **REL_ID=2;3 REL_TYPE=RA**
   [nuclear energy is the mother of the atomic bomb]$_{ADU2}$
   [pesticides that multiplied crops have been banned
   for making food unhealthy]$_{ADU3}$

In example (18), REL_TYPE=RA and REL_ID=1 indicate that ADU2 is an inference of ADU1, whereas in example (19) the REL_TYPE=RA and REL_ID=2;3 are annotated in ADU1 because it is an inference of ADU2 and ADU3, which appear in the original text below ADU1.

- Conflict is the argumentative relationship assigned when two ADUs present contradictory information or when these ADUs contain conflicting or divergent arguments (20). We consider that two ADUs are contradictory 'if they are extremely unlikely to be considered true simultaneously' [27].

  (20a) [vol tenir és dret a formar una família]$_{ADU1}$
     [formar famílies no és un dret]$_{ADU2}$ **REL_ID=1 REL_TYPE=CA**
  (20b) [she wants to have the right to form a familiy]$_{ADU1}$
     [to form families is not a righty]$_{ADU2}$ **REL_ID=1 REL_TYPE=CA**

- Reformulation is the argumentative relationship in which two ADUs have approximately the same or a similar meaning, that is, an ADU reformulates or paraphrases the same discourse argument as that of another ADU (21). The reformulation or paraphrase involves changes at different linguistic levels, for instance, morphological, lexical, syntactic and discourse-based changes [28].

  (21a) [ja n'hi ha prou de paternalism]$_{ADU1}$
     [ja n'hi ha prou que ens tracten com a xiquetes]$_{ADU2}$ **REL_ID=1 REL_TYPE=MA**
  (21b) [enough of paternalism]$_{ADU1}$
     [enough of treating us like children]$_{ADU2}$ **REL_ID=1 REL_TYPE=MA**·

It should be noted that repetitions are not considered reformulations. A repetition contains a claim or statement with the same content as a previous one, that is, the same argument. We consider an ADU to be a repetition only if it is exactly the same as a previous one and we do not therefore segment them and they are not annotated as ADUs.

It is worth noting that, when a team mentions the opposing team's argument, that mention is not considered an argument. When their reasoning is referred to, the reference will be ascribed directly to the opposing team's argument.

### 3.4. Annotation Process

The annotation of the *VivesDebate* corpus was manually performed by two different students of Linguistics specially trained for this task for three months and supervised by two expert annotators (the annotators are members of the Centre de Llenguatge i Computació (CLiC) research group (http://clic.ub.edu/, accessed on 2 August 2021)). The annotation of the corpus was carried out in two main phases. The aim of the first phase was twofold: for the training of the annotators and for defining the annotation guidelines, that is, to establish the definitive tagset and criteria with which to annotate the debates. In this phase, we conducted different Inter-Annotator Agreement tests in order to validate the quality of the annotation of the different tasks involved, that is, the revision of automatic transcriptions, the segmentation of each debate into ADUs and the annotation of argumentative relations between ADUs. These tests allow us to evaluate the reliability of the data annotated, which basically means whether or not the annotators applied the

same criteria for solving the same problem in a consistent way. These inter-annotator tests are also useful for evaluating the quality of the annotation guidelines, that is, to check whether the different types of phenomena to be treated are covered and the criteria are clearly explained, and to update the guidelines when necessary. In the second phase, after the training of the annotators, the remaining files in the corpus were annotated by each annotator independently.

Inter-Annotator Agreement Tests

We carried out, first, a qualitative analysis in order to validate that the team of annotators was applying the same criteria in the revision of the automatic transcriptions of debates. This analysis consisted of the revision of three files (*Debate15.csv*, *Debate11.csv*, and a debate from the previous year's edition, which was not included in our corpus) by the two annotators in parallel and the comparison of the results obtained by the senior annotators. The team met to discuss the problems arising from the comparison of the results in order to resolve doubts and inconsistencies. We devoted three sessions to this until we solved all disagreements and reached the same results in the revision of transcriptions, which explains why we revised the three files selected (one per session). The initial guidelines were updated with the new criteria established, such as following the criteria of the linguistic portal of the Catalan Audiovisual Media Corporation for spelling the named entities or writing down the main ideas in those cases in which the quality of the audio did not allow us to understand part of the message conveyed. In a nutshell, we ensure that the text of the ADUs was transcribed correctly, maintaining the linguistic variant originally used, whereas in the remaining text we applied a more superficial revision.

Once the transcription of texts obtained reliable results, we initiated the segmentation task, which was by far the most difficult task in the whole annotation process. We conducted three Inter-Annotator Agreement (IAA) tests until we reached an acceptable agreement for the segmentation of the transcribed texts into ADUs (see Table 1). We used the same file (*Debate6.csv*) in the two tests. We calculated the observed agreement and the Krippendorff's alpha [29]. The criteria followed for the evaluation of the Inter-Annotator Agreement test were the following:

- In the case of the PHASE, STANCE and argument REL_TYPE tags, we considered agreement to be reached when the annotators assigned the same value to each tag, while disagreement was considered to be when the value was different.
- In the case of the ADU tag, we considered agreement to exist when the span of the ADUs matched exactly, and disagreement to exist when the span did not match at all or coincided partially. We have also conducted a third Inter-Annotator Agreement test for evaluating the ADU tag considering partial agreement. In this case, we considered agreement to exist when the span of the ADUs coincided partially (22)–(25).

(22a) [la cosificació que s'està fent de la dona]$_{ADU1}$
(22b) [the objectivation of women]$_{ADU1}$
(23a) [**ens centrarem en** la cosificació que s'està fent de la dona]$_{ADU1}$
(23b) [**we will focus on** the objectivation of women]$_{ADU1}$
(24a) [el vincle que es genera entre ella i el nadó que porta al seu ventre]$_{ADU1}$ [**és trencat de manera miserable**]$_{ADU2}$
(24b) [the bond between her and the baby she carries in her womb]$_{ADU1}$ [**is broken in a miserable way**]$_{ADU2}$
(25a) [ el vincle que es genera entre ella i el nadó que porta al seu ventre és trencat de manera miserable]$_{ADU1}$
(25b) [the bond between her and the baby she carries in her womb is broken in a miserable way]$_{ADU1}$.

**Table 1.** Results of the Inter-Annotator Agreement Tests.

| Tag | Observed Agreement % | Krippendorff's Alpha |
|---|---|---|
| STANCE (AGAINST/FAVOUR) | 99.05 | 0.979 |
| PHASE (INTRO/ARG1,ARG2,ARG3/CONC) | 94.60 | 0.925 |
| REL_TYPE (RA/CA/MA) | 86.00 | 0.913 |
| ADU (1st IAA Test) | 70.80 | 0.392 |
| ADU (2nd IAA Test) | 76.60 | 0.777 |
| ADU (3rd IAA Test partial disagreements) | 91.20 | 0.917 |

The disagreements found are basically of two types: (a) the inclusion or omission of words at the beginning or at the end of the ADU (22) vs. (23); and (b) the segmentation of the same text into two ADUs or a single ADU (24) vs. (25), the latter being stronger disagreement than the former. For instance, one of the annotators considered 'is broken in a miserable way' to be a different ADU (24), whereas the other annotator considered this segment part of the same ADU (25). Finally, we agreed that it should be annotated as a single ADU (25), because 'is broken' is the main verb of the sentence, and the argument is that what is broken is the bond between the mother and the baby.

As shown in Table 1, the results obtained in the Inter-Annotator Agreement tests for the PHASE, STANCE and REL_TYPE tags are almost perfect (above 0.97), and are acceptable for the ADU tags (0.77), which correspond to the segmentation into ADUs and the assignation of the type of argument, following Krippendorff [29] recommendations. The observed agreement (91.20%) and the corresponding alpha value (0.91) for the ADU tag increase when we consider there to be partial agreement ($\alpha \geq 0.80$ the customary requirement according to Krippendorff). The team of annotators met once a week to discuss problematic cases and resolve doubts in order to minimise inconsistencies and guarantee the quality of the final annotation. The results obtained are very good given the complexity of the task.

## 4. The *VivesDebate* Corpus

### 4.1. Data Collection

The *VivesDebate* corpus has been created from the transcripts of the 29 complete debates carried out in the framework of the 2019 *"Xarxa Vives d'universitats"* university debate tournament. During this competition, 16 different teams from universities belonging to the autonomous regions of Valencia, Catalonia and the Balearic islands debated in the Catalan language on the topic *"Should surrogacy be legalised?"* In addition to the original language of the annotated data, automatic translations to Spanish and English languages using the MLLP machine translation toolkit [30,31] have also been included. The results and the evaluation of the debates were directly retrieved from the organisation, but were post-processed by us in order to focus on the argumentative aspects of the debates and to preserve the anonymity of the jury and the participant teams. Furthermore, we would also like to remark that the data collected is part of a competition where the stances (i.e., favour or against) are assigned randomly. Thus, any argument or opinion existing in the corpus is used to elaborate a logically solid reasoning, but it is not necessarily supported by the participants.

### 4.2. Structure and Properties

The *VivesDebate* corpus is structured into 30 different CSV documents publicly available in Zenodo (https://doi.org/10.5281/zenodo.5145656, accessed on 2 August 2021). The first 29 documents each correspond to a unique debate, containing three phases: introduction, argumentation and conclusion. The structure of each document (Table 2) contains the identified ADUs (rows) in Catalan (ADU_CAT), Spanish (ADU_ES), and English (ADU_EN), and covers the six features that define every identified ADU (columns). First of all, each ADU is assigned a unique ID created following the chronological order (i.e., 1, 2, . . . , *N*; where *N* is the total number of ADUs in a debate). This ID allows for an intuitive

representation of the flow of discourse in each debate. Second, each ADU is classified into one of the three phases of competitive debate (i.e., *INTRO*, *ARG1*, *ARG2*, and *CONC*) depending on when has it been uttered. Third, each ADU that forms part of one of the arguments put forward by the debaters is assigned an argument number. This number allows the grouping of every identified ADU under the same claim. The same number used for the ADUs belonging to different stances does not imply any type of relation between them, since their related claim is different (i.e., argument number 1 in favour and argument number 1 against stand for two different arguments). Fourth, each ADU is classified according to the stance (i.e., in favour or against) for which it was used. Finally, the existing argumentative relations between ADUs are identified and represented with the relation type (i.e., Conflicts or CA, Inferences or RA, and Rephrases or MA), and the ID(s) of the related ADU(s).

**Table 2.** Structure of the *VivesDebate* corpus CSV documents (*Debate7.csv*). (*) An empty value in the *Arg. Number* column indicates that the ADU does not explicitly belong to any argument presented by the Favour or Against team to specifically support their stance.

| ID | Phase | Arg. Number (*) | Stance | ADU_CAT, ADU_ES, ADU_EN | Related ID | Relation Type |
|----|-------|------------------|--------|--------------------------|------------|----------------|
| 1 | INTRO | | FAVOUR | *quan mireu aquí què veieu* (*cuando miráis aquí qué veis*) (*when you look here what do you see*) | | |
| 2 | INTRO | | FAVOUR | *cinquanta euros* (*cincuenta euros*) (*fifty euros*) | 1 | RA |
| 3 | INTRO | | FAVOUR | *el nostre nou déu* (*nuestro dios*) (*our new god*) | 2 | RA |
| … | … | … | … | … | … | … |
| 43 | ARG1 | 1 | FAVOUR | *vivim en un món on ha guanyat els valors del neoliberalisme* (*vivimos en un mundo dominado por los valores del neoliberalismo*) (*we live in a world dominated by the values of neoliberalism*) | | |
| 44 | ARG1 | 1 | FAVOUR | *uns valors que ens diuen que si tenim diners som guanyadors* (*valores que dicen que si tenemos dinero somos ganadores*) (*values that say that if we have money we are winners*) | 43 | RA |
| 45 | ARG1 | 1 | FAVOUR | *i si som guanyadors podem comprar tot allò que desitgem* (*y si somos ganadores podemos comprar lo que deseemos*) (*and if we are winners we can buy whatever we want*) | 44 | RA |
| 46 | ARG1 | 1 | FAVOUR | *és el model que està imperant en la gestació subrogada* (*es el modelo que impera en la gestación subrogada*) (*is the prevailing surrogacy model*) | 43; 44; 45 | RA |
| … | … | … | … | … | … | … |
| 144 | ARG1 | 3 | AGAINST | *per suposat que no* (*por supuesto que no*) (*of course not*) | 143 | CA |
| 145 | ARG1 | 3 | AGAINST | *no és que vullguem que hi haja més xiquets* (*no es que queramos que haya más niños*) (*not that we want there to be more children*) | 140 | RA |
| 146 | ARG1 | 3 | AGAINST | *sinó tot el contrari* (*sino todo lo contrario*) (*quite the contrary*) | 145 | MA |
| … | … | … | … | … | … | … |

Each document (i.e., debate) was annotated independently and in its entirety. The conclusions of each team are considered to be a unique ADU separately, since they represent a good summary of the argumentative discourse from both teams' perspectives. In addition to the 29 annotated debate documents, the corpus has a supplementary evaluation file (*VivesDebate_eval.csv*). The Jury's evaluation of one debate (*Debate29.csv*) was not available for the creation of the evaluation file. Thus, this file contains an anonymised version of

the Jury evaluation of the first 28 debates (Table 3). However, since some aspects used by the judges to evaluate the debating teams are not reflected in our corpus (e.g., oral fluency, grammatical correctness, and non-verbal language), we have excluded them from our argumentative evaluation file. A numerical score in the range of 0 to 5 (100%), which combines the thesis solidity (35%), the argumentation quality (35%), and the argument adaptability (30%) is provided as the formal evaluation for each debate. The presented structure makes the *VivesDebate* corpus a very versatile resource for computational argumentation research. Not only because of its size, but due to all the information it contains, it can be used for Argument Mining, Argument Analysis and representation, Argument Evaluation and argument summarising (i.e., generation) research tasks.

**Table 3.** Structure of the *VivesDebate* evaluation file.

| Debate | Stance | Score | Thesis Solidity | Argument Quality | Adaptability |
|---|---|---|---|---|---|
| Debate1 | Favour | 3.32 | 3.25 | 3.37 | 3.33 |
| Debate1 | Against | 3.29 | 3.33 | 3.18 | 3.38 |
| Debate2 | Favour | 3.41 | 3.5 | 3.33 | 3.42 |
| Debate2 | Against | 3.43 | 3.17 | 3.58 | 3.58 |
| . . . | . . . | . . . | . . . | . . . | . . . |
| . . . | . . . | . . . | . . . | . . . | . . . |
| Debate28 | Favour | 2.75 | 3.25 | 2.75 | 2.17 |
| Debate28 | Against | 2.36 | 2.77 | 2.25 | 2.00 |

The resulting *VivesDebate* corpus comprises a total of *139,756* words (tokens). Furthermore, the corpus presents an average of *4819* words per document independently annotated. Words are grouped into a total of *7810* ADUs, with an average of *269* ADUs per document. In our corpus, an argument is built from multiple ADUs sharing argumentative relations. A total of *1558* conflicts, *12,653* inferences, and *747* rephrases between the identified ADUs have been annotated in the *VivesDebate* corpus, with an average of *54* conflicts, *436* inferences, and *26* rephrases per document. A summary of the structure and a breakdown for each of the included debates is presented in Table 4. In addition to all these corpus statistics, we retake the "argument density" metric proposed in [32]. This metric computes the density of arguments in a corpus by normalising the number of annotated inference relations to the total word count. The *VivesDebate* presents an "argument density" of 0.091, which is significantly higher compared to the densities of previously existing similar corpora such as the US2016 [32] (0.028 density) for *97,999* words, or the DMC [33] (0.033 density) for *39,694* words.

**Table 4.** Structure and properties of the *VivesDebate* corpus. Score F and A represent the score assigned to the favour and against teams respectively according to our processing of the original evaluation.

| File | Words | ADUs | Conflicts | Inferences | Rephrases | Score F | Score A |
|---|---|---|---|---|---|---|---|
| *Debate1.csv* | 3979 | 198 | 4 | 158 | 22 | 3.32 | 3.29 |
| *Debate2.csv* | 5178 | 371 | 60 | 310 | 32 | 3.41 | 3.43 |
| *Debate3.csv* | 4932 | 311 | 63 | 360 | 22 | 4.39 | 4.31 |
| *Debate4.csv* | 6243 | 308 | 8 | 229 | 37 | 4.01 | 4.15 |
| *Debate5.csv* | 5389 | 270 | 45 | 505 | 48 | 4.38 | 3.28 |
| *Debate6.csv* | 4387 | 324 | 45 | 219 | 42 | 2.94 | 3.02 |
| *Debate7.csv* | 4523 | 299 | 11 | 236 | 18 | 3.31 | 3.13 |
| *Debate8.csv* | 4933 | 220 | 5 | 185 | 15 | 3.31 | 3.92 |
| *Debate9.csv* | 5574 | 352 | 45 | 309 | 18 | 4.12 | 4.21 |
| *Debate10.csv* | 4284 | 279 | 12 | 207 | 39 | 4.39 | 3.46 |
| *Debate11.csv* | 5720 | 239 | 5 | 202 | 16 | 3.60 | 3.64 |
| *Debate12.csv* | 5305 | 283 | 83 | 477 | 49 | 4.12 | 4.36 |
| *Debate13.csv* | 3646 | 138 | 10 | 106 | 6 | 2.94 | 2.60 |

**Table 4.** *Cont.*

| File | Words | ADUs | Conflicts | Inferences | Rephrases | Score F | Score A |
|------|-------|------|-----------|------------|-----------|---------|---------|
| *Debate14.csv* | 4790 | 302 | 74 | 400 | 47 | 3.83 | 3.80 |
| *Debate15.csv* | 4550 | 173 | 23 | 113 | 13 | 3.95 | 3.94 |
| *Debate16.csv* | 4887 | 288 | 94 | 639 | 53 | 3.33 | 3.39 |
| *Debate17.csv* | 3891 | 164 | 8 | 123 | 8 | 3.00 | 3.26 |
| *Debate18.csv* | 3701 | 166 | 6 | 149 | 4 | 2.80 | 2.77 |
| *Debate19.csv* | 4645 | 186 | 13 | 159 | 1 | 4.24 | 4.34 |
| *Debate20.csv* | 5484 | 306 | 33 | 1306 | 55 | 3.53 | 3.49 |
| *Debate21.csv* | 5064 | 278 | 102 | 1076 | 42 | 3.17 | 3.18 |
| *Debate22.csv* | 4669 | 330 | 16 | 408 | 1 | 4.40 | 4.22 |
| *Debate23.csv* | 4420 | 266 | 136 | 917 | 26 | 2.74 | 2.69 |
| *Debate24.csv* | 5139 | 267 | 380 | 1002 | 39 | 4.41 | 4.37 |
| *Debate25.csv* | 4828 | 321 | 7 | 337 | 0 | 4.09 | 3.88 |
| *Debate26.csv* | 4440 | 290 | 16 | 328 | 5 | 4.16 | 3.93 |
| *Debate27.csv* | 5012 | 234 | 106 | 645 | 24 | 3.49 | 2.33 |
| *Debate28.csv* | 4254 | 310 | 21 | 344 | 2 | 2.75 | 2.36 |
| *Debate29.csv* | 5889 | 337 | 51 | 1203 | 72 | - | - |
| *VivesDebate* | 139,756 | 7810 | 1558 | 12,653 | 747 | - | - |

## 5. Related Work: Other Computational Argumentation Corpora

As noted in the introduction, the existing resources for computational argumentation present significant differences depending on their main purposes. Thus, we consider it important to contextualise our contribution to the computational argumentation research within the existing related work. For that purpose, we present a thorough comparison between the most prominent available resources for the computational argumentation research community. One of the first public corpora focused on the Argument Mining task was presented in [34], where the authors annotated 90 persuasive essays in English obtained from an online forum. In this corpus, two different aspects of arguments were annotated, the argument components (i.e., claim and premise) and the argumentative relations (i.e, attack and support). Another early resource to satisfy the needs of Argument Mining researchers was presented in [35]. The authors present a new corpus of 112 annotated "microtexts", short and dense written arguments in German, which were also professionally translated into English. This corpus was annotated taking into account the argumentative structure of the text, where each argument has a central claim with an argumentative role (i.e., proponent and opponent), and several elements with different argumentative functions (i.e., support, attack, linked premises and central claims). These "microtexts" were generated in a controlled experiment where 23 participants were instructed to write argumentative text on a specific topic. A different approach was introduced in [33], where dialogue spoken argumentation samples were used to create the *Dispute Mediation Corpus* (DMC). Three different sources were considered to retrieve up to 129 mediation excerpts, which were analysed by a unique professional annotator. The sources from which these excerpts were annotated were 58 transcripts found in academic papers, 29 online website mediation scripts, 14 scripts provided by professional mediators, and 28 analyses of meta-discourse elements in mediation interactions from a mixture of the previous sources. The DMC corpus was annotated using the Inference Anchoring Theory (IAT), containing up to eleven structural features of arguments useful for the Argument Mining task: locutions, assertions, assertive questions, pure questions, rhetorical questions, assertive challenges, pure challenges, popular concessions, inferences, conflicts and rephrases. Furthermore, graphical representations of the complete structures useful for Argument Analysis can be loaded in the OVA+ (http://ova.arg-tech.org/, accessed on 2 August 2021) tool. In addition to the Argument Mining and analysis tasks, the automatic evaluation of arguments is an important aspect in the analysis of argumentative discourses. Reference [36] presents the *Consumer Debt Collection Practices* (CDCP) corpus, where 731 user comments from an online

forum are annotated with their argumentative structures, capturing the *strength* of the identified arguments. In the CDCP corpus, each comment is segmented into elementary units (i.e., Facts, Testimony, Value, Policy and Reference). Support relations between these elementary units are annotated in order to provide structural information. The authors defined the evaluability of an argument for those cases in which all the propositions that make up this argument are supported by an explicit premise of the same type of elementary unit. The *strength* of an argument is measured by comparing the type of the elementary units that comprise it. Another important part of computational argumentation, which was not approachable from the reviewed corpora, is the automatic generation of natural language arguments. This is a recent research topic, which requires an important amount of data to achieve competitive results. In [37], the authors present a new annotated corpus aimed at approaching this task. The GPR-KB-55 contains 200 speeches from a debate competition that were analysed, each one debating one of the 50 different topics existing in these speeches. The resulting corpus consists of 12,431 argument pairs containing a claim and its rebuttal with annotations regarding the relevance of a claim to its motion, the stance of the claim, and its appearance in a piece of speech (i.e., mentioned/not mentioned, explicit/implicit). Even though some linguistic annotations were done, the argument structure or the flow of discourse were not annotated in the GPR-KB-55 corpus. This corpus is part of the IBM Project Debater (https://www.research.ibm.com/artificial-intelligence/project-debater/, accessed on 2 August 2021), which encompasses a large set of different corpora, each one aimed at a specific task of the argumentative process. A different perspective on natural language Argument Generation is presented in [38], where the authors provide a new corpus aimed at the word-level summarisation of arguments. The DebateSum corpus consists of 187,386 debate summaries without any structural annotation, retrieved from the debate tournaments organised by the National Speech and Debate Association. The only annotation provided by this corpus is the segmentation of arguments–evidences–summary triplets extracted directly from the transcripts. Again, the usefulness of this resource remains strongly linked to the specific tasks of argument summarisation and language modelling. At this point, it is possible to observe the strong dependence between the analysed corpora and the different tasks of computational argumentation. The *US2016* debate corpus was presented in [32] as the largest argumentative corpus with great versatility between different aspects of argumentation such as discourse analysis, Argument Mining, and automatic Argument Analysis. The *US2016* compiles the transcripts of the 2016 US presidential election TV debate and the subsequent online forum debate (i.e., Reddit). The text is analysed and divided into argument maps consisting of 500 to 1500 words. The annotation process is carried out independently for each argument map, where the text is segmented into ADUs and argumentative relations between ADUs are identified. The final corpus has 97,999 words, with a sub-corpus of 58,900 words from the TV debates and 39,099 words from the Reddit discussion. Despite the improvement achieved with this new corpus, we still identify two major issues that may hinder the performance of the trained models and the scope of the experiments: the quality of the uttered arguments, and the traceability of discourse. Electoral campaigns and debates are usually focused on reaching the majority of voters rather than properly using arguments, or having a rational debate. Furthermore, the arguments retrieved from an online forum might neither be of the ideal quality. Thus, the trained models using this data can be biased in a way that does not reflect the reality of a more rational and logical argumentation. Finally, the most recent argumentative corpus was presented in [39]. The authors present the ReCAP corpus of monologue argument graphs extracted from German education politics. More than 100 argument graphs are annotated from natural language text sources like party press releases and parliamentary motions. This corpus annotates the ADU segments identified in the text and relations between different ADUs (i.e., inferences). Furthermore, the authors have also included annotations of the underlying reasoning pattern (i.e., argumentation schemes) of arguments. These are not the unique resources published in the literature for computational argumentation research. Many research done in Argument Mining includes

new corpora, for the healthcare domain in [40], for legal argumentation in [41], and for online social network analysis in [42] among other different domains. However, for our comparison, we have focused on the most used corpora in computational argumentation research, and corpora created from a more generalist perspective.

A comparison of the previously analysed corpora is presented in Table 5. Furthermore, we have added the *VivesDebate* corpus to the comparison in order to provide a reference to understand the significance of our contribution. Seven different features that we consider indicators of the quality of a corpus have been analysed in our comparison. First, the format of the argumentative data indicates if the arguments are retrieved from a monologue (M) or a dialogue (D). Furthermore, it is also important to know the source of the arguments, if they come from a text source (T) or from a speech transcript (S). The domain indicates the context from which the corpus has been created (e.g., competitive debate, online forum, etc.). This is a key feature to determine the quality of the arguments contained in the resulting corpus, since major linguistic aspects, such as the richness of vocabulary or the originality of arguments, will be significantly different from one domain to another. The tasks feature indicates in which argumentative tasks a corpus can be useful: Argument Mining (AM), Argument Analysis (AA) and representation, Argument Evaluation (AE), Argument Generation (AG), and Argument Summarisation (AS). This feature is important to observe the versatility of each analysed corpus. The language indicates if a corpus is available in English (EN), German (DE), or Catalan (CAT). Finally, we have taken into account the size of each corpus in words (W) and/or sentences (S); and the annotation ratio, which indicates the average number of words (or sentences) per each independently annotated document w/d (s/d). This last feature can give us an idea of the contextual information preserved in the annotation process. For instance, it is not the same to annotate a complete debate (higher annotation ratio), than to split the debate into smaller argumentative structures to simplify the annotation process (lower annotation ratio).

**Table 5.** Comparison of computational argumentation corpora. (*) Automatically translated languages.

| Research | Identifier | Format | Source | Domain | Tasks | Language | Size | Annotation Ratio |
|---|---|---|---|---|---|---|---|---|
| [34] | Persuasive Essays | M | T | Online Forum | AM | EN | 34,917 (W) | 388 w/d |
| [35] | Microtexts | M | T | Controlled Experiment | AM | EN + DE | 576 (S) | 5 s/d |
| [33] | DMC | D | S | Academic + Online + Professional | AM + AA | EN | 18,628 (W) | 144 w/d |
| [36] | CDCP | D | T | Online Forum | AM + AE | EN | 4931 (S) | 6.7 s/d |
| [37] | GPR-KB-55 | D | S | Competitive Debate | AG | EN | 12,431 (S) | 41 w/d |
| [32] | US2016 | D | T + S | Political + Online | AM + AA | EN | 97,999 (W) | 189 w/d |
| [32] | US2016TV | D | S | Political | AM + AA | EN | 58,900 (W) | 492 w/d |
| [32] | US2016Reddit | D | T | Online Forum | AM + AA | EN | 39,099 (W) | 137 w/d |
| [38] | DebateSum | D | S | Competitive Debate | AS | EN | 101M (W) | 520 w/d |
| [39] | ReCAP | M | T | Political | AM + AA | DE + *EN*(*) | 16,700 (W) | 150 w/d |
| | *VivesDebate* | D | S | Competitive Debate | AM + AA + AE + AG/AS | CAT + *ES*(*) + *EN*(*) | 139,756 (W) | 4819 w/d |

Thus, it is possible to observe how, in addition to the quality improvement mainly due to the source (i.e., competitive debate), our corpus can be useful in a wider variety of computational argumentation tasks. Furthermore, the *VivesDebate* corpus presents an annotation ratio that is significantly higher compared to the previous work. This approach makes it possible to improve the richness of the annotations by keeping longer-term argumentative relations and allowing a complete representation of the flow of the debate.

## 6. Conclusions

In this paper, we describe *VivesDebate*, a new annotated multilingual corpus of argumentation created from debate tournament transcripts. This work represents a major step forward in publicly available resources for computational argumentation research. Next, we summarise the main improvements brought about by the creation of this corpus.

First, because of its size. The *VivesDebate* corpus is, to the best of our knowledge, one of the largest publicly available resources annotated with relevant argumentative propositions, and argumentative and dialogical relations. With a total of *139,756* words and

an argument density of 0.091, in addition to its size, the *VivesDebate* corpus also improves the previously available argumentation corpora in terms of their density.

Second, because of the quality of the argumentative reasoning data. The *VivesDebate* corpus has been created from the transcripts of 29 complete competitive debates. Annotating spoken argumentation is usually harder and more expensive than textual argumentation, so the majority of the publicly available corpora for computational argumentation research are created from social networks and online forum debates. Furthermore, most of the available spoken argumentation corpora are from the political debate domain, which does not have a solid structure and the quality of argumentation is harder to evaluate. By creating a new corpus from the transcripts of a debate tournament, the improvement of the argumentative quality compared to previously available corpora is threefold: (i) debate tournaments have a well-defined argumentation structure, which eases their modelling; (ii) the only motivation behind the debates is the argumentation itself, so that participants need to argue using the strongest arguments and present a coherent reasoning to win the debate; and (iii) the debates are objectively evaluated by an impartial jury, analysing parameters that are directly related to the quality of arguments and argumentation.

Third, because of its versatility. The size, the structure, the annotations, and the content of the *VivesDebate* corpus make it useful for a wide range of argumentative tasks such as argumentative language modelling, the automatic identification of ADUs in an argumentative dialogue (i.e., Argument Mining), the elaboration and analysis of complex argument graphs (i.e., Argument Analysis), the automatic evaluation of arguments and argumentative reasoning (i.e., Argument Evaluation), and the automatic generation of argument summaries (i.e., Argument Generation/Argument Summarising). Furthermore, the corpus is available in its original version in Catalan, and in machine-translated versions to Spanish and English languages, leaving an open door to multilingual computational argumentation research.

Even though the *VivesDebate* corpus provides significant improvements over existing resources for computational argumentation research, it has its own limitations. The debates contained in the corpus belong to a unique tournament, which means that every annotated debate will have the same topic in common. This feature is directly related to the observable language distribution, which will be biased by the *"Should surrogacy be legalised?"* topic. However, since our corpus is aimed at computational argumentation research rather than language modelling, this should not be an important issue. Furthermore, this data bias can be easily amended with a topic extension of the *VivesDebate* corpus. The other main limitations of the corpus are the Spanish and English machine-translated versions, which may not be as linguistically correct as the original version in Catalan.

As future work, we plan to overcome some of these limitations and to deepen the argumentative analysis and annotation of the corpus. First, we plan to improve the Spanish and English machine-translated versions of the *VivesDebate* corpus with a professional translation. We also want to improve the argumentative annotations of the corpus by deepening the logical and rational aspects of argumentation. In its current form, it is possible to perform a general structural analysis of the arguments. With the identification and annotation of stereotyped patterns of human reasoning (i.e., argumentation schemes [3]), it will be possible to bring the automatic detection and analysis of arguments to a deeper level. However, this is a complex task, and it has only been superficially researched in the literature. Finally, we are also exploring the possibility of organising a new shared task focused on the argumentative analysis of natural language inputs.

**Author Contributions:** Conceptualization, R.R.-D., S.H. and A.G.-F.; Data curation, M.N.; Formal analysis, R.R.-D.; Funding acquisition, A.G.-F.; Methodology, M.N. and M.T.; Project administration, M.T. and A.G.-F.; Resources, R.R.-D., M.N., M.T., S.H. and A.G.-F.; Supervision, S.H. and A.G.-F.; Validation, M.N. and M.T.; Writing—original draft, R.R.-D. and M.T.; Writing—review & editing, R.R.-D., M.N., M.T., S.H. and A.G.-F. All authors have read and agreed to the published version of the manuscript.

**Funding:** This research was funded by the Spanish Government project PID2020-113416RB-I00; the Valencian Government project PROMETEO/2018/002; the MISMIS-Language project (PGC2018-096212-B-C33) funded by Ministerio de Ciencia, Innovación y Universidades; and the CLiC Research Group (2017SGR341) funded by Generalitat de Catalunya.

**Institutional Review Board Statement:** Not applicable.

**Informed Consent Statement:** Not applicable.

**Data Availability Statement:** The *VivesDebate* corpus is freely available to download from Zenodo (https://doi.org/10.5281/zenodo.5145656, accessed on 2 August 2021) under a Creative Commons Attribution-NonCommercial-ShareAlike 4.0 International license (CC BY-NC-SA 4.0).

**Acknowledgments:** We gratefully acknowledge the *Xarxa Vives d'Universitats* for organising the debate tournaments, making publicly available the videos, and sharing the evaluations with the research community.

**Conflicts of Interest:** The authors declare no conflict of interest.

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
