# Peer review of "VivesDebate: A New Annotated Multilingual Corpus of Argumentation in a Debate Tournament"

_applsci, doi:10.3390/app11157160_

Round 1

Reviewer 1 Report

This paper presented a VivesDebate, which is a new annotated multilingual corpus of argumentation in a debate tournament. The reviewer thinks that VivesDebate is a valuable resource for computational argumentation research. In general, this paper is well-written, well-organized, and easy to follow. Please see the attached file for comments.

Author Response

Dear reviewer and editor, we would like to personally thank you for the reviews that we received. By following your suggestions, we have been able to improve our work. Therefore, please receive our most sincere thanks.

1.- The rule-of-thumb applied to the segmentation of ADUs was to ensure the comprehensibility of each ADU, i.e. to ensure that the meaning conveyed is easy to interpret. The ADUs are generally sentences or part of a complex sentence (coordinated and subordinated clauses), which are the cases exemplified in the different criteria applied.

We have added in 244 line the following: The ADUs are generally equivalent to a sentence or a dependent clause (for instance a subordinated or coordinated clause).

2.- The Inter-Annotator Agreement tests are applied specifically in the training phase of the annotators with a twofold aim: a) to improve and update, when necessary, the annotation guidelines and b) to ensure that annotators are consistently applying the same criteria. Therefore, until we reached an acceptable agreement for the segmentation of texts into ADUs, the annotators did not begin to segment the transcribed texts individually, that is, separately.

We have added in 543 line the following: The team of annotators met once a week to discuss problematic cases and resolve doubts in order to minimise inconsistencies and guarantee the quality of the final annotation.

3.- The VivesDebate corpus has a total of 139,756 words. It is true that the number of "useful" words for each different computational argumentation task may vary. However, in our paper, we present the corpus as a resource for research, and it has not been prepared for any specific task as it would be in a competition or a shared task. Thus, the number of words for every of the mentioned tasks is the same, 139,756 words. The variations on this number will depend on the pre-processing done by the researchers using it for specific argumentation tasks. The number of words may even be different for the same task if the pre-processing is carried out taking into account different aspects of natural language and argumentation.

Reviewer 2 Report

The authors (you) did a great job. The scientific contribution is manifested in setting clear guidelines for the future corpus of argumentation creations. The only thing I can point out is to modify the paper in such a way as to describe in more detail the papers numbered from [33] to [43], in the section "Related Work: Other Computational Argumentation Corpora". 

Author Response

Dear reviewer, we would like to personally thank you for the reviews that we received. By following your suggestions, we have been able to improve our work. Therefore, please receive our most sincere thanks.

In Section 5, we analysed seven different features to compare the previously existing argumentation corpora: the format (i.e., monologue/dialogue), the source (i.e., textual/speech), the domain from which argumentation is annotated, the task that can be addressed using the corpus, the language availability, the size in words or sentences, and the annotation ratio to measure the depth in which each corpus has been annotated. We consider that these features allow the reader to fully picture each of the compared corpora. However, we have strengthened the descriptions of some of these corpora in the text:

(line 635) "A different approach was introduced in \cite{janier2016corpus}, where dialogue spoken argumentation samples were used to create the \textit{Dispute Mediation Corpus} (DMC). Three different sources were considered to retrieve up to 129 mediation excerpts which were analysed by a unique professional annotator. The sources from which these excerpts were annotated are 58 transcripts found in academic papers, 29 online website mediation scripts, 14 scripts provided by professional mediators, and 28 analyses of meta-discourse elements in mediation interactions from a mixture of the previous sources. The DMC corpus was annotated using the Inference Anchoring Theory (IAT), containing up to eleven structural features of arguments useful for the argument mining task: locutions, assertions, assertive questions, pure questions, rhetorical questions, assertive challenges, pure challenges, popular concessions, inferences, conflicts, and rephrases. Furthermore, graphical representations of the complete structures useful for argument analysis can be loaded in the OVA+\footnote{\url{http://ova.arg-tech.org/}} tool."

(line 651) "In the CDCP corpus, each comment is segmented into elementary units (i.e., Facts, Testimony, Value, Policy, and Reference). Support relations between these elementary units are annotated in order to provide structural information. The authors defined the evaluability of an argument for those cases in which all the propositions that make up this argument are supported by an explicit premise of the same type of elementary unit. The \textit{strength} of an argument is measured by comparing the type of the elementary units that make it up."

(line 661) "In \cite{orbach2019dataset}, the authors present a new annotated corpus aimed at approaching this task. The GPR-KB-55 contains 200 speeches from a debate competition that were analysed, each one debating one of the 50 different topics existing in these speeches. The resulting corpus consists of 12431 argument pairs containing a claim and its rebuttal with annotations regarding the relevance of a claim to its motion, the stance of the claim, and its appearance in a piece of speech (i.e., mentioned/not mentioned, explicit/implicit). Even though some linguistic annotations were done, the argument structure or the flow of discourse were not annotated in the GPR-KB-55 corpus."

(line 672) "The DebateSum corpus consists of 187,386 debate summaries without any structural annotation, retrieved from the debate tournaments organised by the National Speech and Debate Association. The only annotation provided by this corpus is the segmentation of arguments-evidences-summary triplets extracted directly from the transcripts."

Reviewer 3 Report

This manuscript presents an annotated multilingual corpus for computational argumentation. The manuscript is well-written and the subject is interesting however there are some points that must be revised to make this manuscript publishable:

1- In a scientific text, exaggerative language should be avoided to allow the audience to judge the quality of the presented methodology based on their studies of the literature. Hence, the phrases like "the largest", "the richest", and "most versatile professional debate ..." have to be removed while describing the presented VivesDebate.

2- Although the introduction is well-written, I suggest adding the general and specific objectives of the study as bullet-points at the end of the introduction section to clarify it for the audience.

3- A descriptive section should be added to focus on the deep learning model used, its architecture and related hyperparameters, the training process, and methods to improve this algorithm. 

4- Although removing punctuation marks can be helpful to provide a good translation from spanish to english, but it could sometimes change the meaning of the sentence (ex. imperative could be translated wrongly in present tense). Hence, in the segmentation process section, it should be elaborated that for what reason exactly the punctuation mark removal is considered.

In general, I believe this is an interesting subject however more examples could be provided that show the deficiencies of the system also, and not only the strengths of the proposed system.

Author Response

Dear reviewer, we would like to personally thank you for the reviews that we received. By following your suggestions, we have been able to improve our work. Therefore, please receive our most sincere thanks.

1.- In the abstract, we have removed the superlatives (line 5):

"In this paper, we present VivesDebate, a large, richly annotated, and versatile professional debate corpus for computational argumentation research."

We have also removed these phrases from the corpus description section. Furthermore, we have relaxed the use of such expressions in the conclusion, after doing the comparison between existing argumentative corpora (that can support our statements):

(line 745) "The VivesDebate corpus is, to the best of our knowledge, one of the largest publicly available resources"

(line 747) "With a total of 139,756 words and an argument density of 0.091, in addition to its size, the VivesDebate corpus also improves the previously available argumentation corpora in terms of their density."

2.- We have considered this option to outline the introduction with bullet points. However, our manuscript is mainly descriptive in nature, and would not be as informative as it would be in a study. Since we are presenting a new corpus, in the introduction we have included a brief paragraph enlightening the most relevant descriptive features of it. We have emphasised the main objective and contributions of the paper in line 79:

"The main objective of this article is to present \textit{VivesDebate}, a new annotated argumentative multilingual corpus from debate tournaments. For that purpose, the contribution of this paper is threefold: (i) the creation of a new resource for computational argumentation research; (ii) the description of the annotation guidelines followed in the creation of the corpus; and (iii) the comparison and review of ten of the most relevant corpora for computational argumentation research. The \textit{VivesDebate} corpus has been created based on three main aspects that are of paramount importance..."

3.- In our paper, we are presenting VivesDebate, a multilingual corpus of argumentation in a debate tournament. The contributions of our paper are the own creation of this new corpus, the thorough description of the annotation guidelines, and a comparison with previously annotated corpora. However, we have not used any deep learning system to carry out any of these tasks. Thus, we believe that the descriptive section mentioned above would not fit in with the contributions presented in our paper.

4.- The transcriptions of each debate automatically obtained by the MLLP transcription system do not contain punctuation marks.  Although including punctuation marks can significantly improve the readability of the transcribed texts, getting wrong punctuation can be worse than providing no punctuation (https://www.theverge.com/2020/2/7/21127852/google-speech-voice-to-text-ai-punctuation). This can be explained because adding punctuation in spoken language is an arbitrary task, and therefore it is difficult to get good results.

We finally decided that it was better not to add punctuation marks, and when the lack of punctuation led to ambiguity the annotators could then view the debate recording to avoid a misinterpretation of the message. 

We have changed this sentence (line 249): “Punctuation marks must be removed from the content of the ADUs”  to “Punctuation marks must not be added to the content of the ADUs. The annotators could view the debate recording to solve ambiguous interpretations and avoid a misinterpretation of the message.